# Teaching Humans Subtle Differences with *DIFF*usion

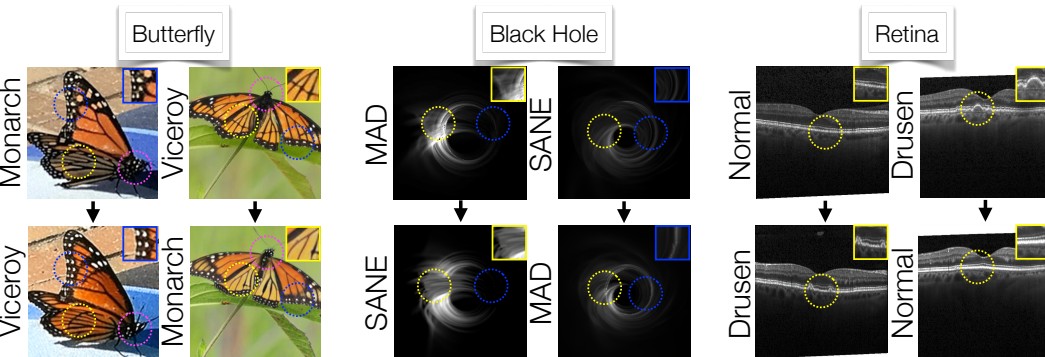

Figure 1: ***DIFF*usion Counterfactuals.** We illustrate the counterfactual results from our methods on the Butterfly dataset, the Black Hole dataset, and the Retina dataset. In the Butterfly dataset, the Viceroy has a cross-sectional line (yellow), a smaller head with less dots (magenta), and more "scaley" dots (blue), compared to the Monarch. In the Black Hole dataset, SANE has more uniform wisps (yellow) and less of a prominent photon ring (blue) as compared to MAD, with these distinguishing features discovered through our method rather than known a priori. In the Retina dataset, normal retinas lack the horizontal line bumps (yellow) present in retinas with drusen.

## Abstract

Scientific expertise often requires recognizing subtle visual differences that remain challenging to articulate even for domain experts. We present a system that leverages generative models to automatically discover and visualize minimal discriminative features between categories while preserving instance identity. Our method generates counterfactual visualizations with subtle, targeted transformations between classes, performing well even in domains where data is sparse, examples are unpaired, and category boundaries resist verbal description. Experiments across six domains, including black hole simulations, butterfly taxonomy, and medical imaging, demonstrate accurate transitions with limited training data, highlighting both established discriminative features and novel subtle distinctions that measurably improved category differentiation. User studies confirm our generated counterfactuals significantly outperform traditional approaches in teaching people to correctly differentiate between fine-grained classes, showing the potential of generative models to advance human visual learning and scientific research.

## 1 Introduction

Generative models, especially large-scale image diffusion models, have transformed text-to-image creation, opening new ways to visualize concepts across various domains. While these models excel in everyday contexts with clear category distinctions, a far more challenging frontier exists in scientific fields where visual differences between categories are so subtle that they often remain unknown and unidentified even to domain experts.

In specialized scientific domains, the complete set of visual features distinguishing between categories may be partially or entirely undiscovered. For example, astronomers studying black hole sim-

ulations have no established verbal characteristics to differentiate MAD from SANE models because these distinguishing features have not yet been comprehensively identified. Entomologists may differentiate Viceroy and Monarch butterflies through the Viceroy's characteristic cross-sectional black line, yet may miss other distinguishing features that could further help the differentiation. This represents the fundamental challenge for visual expertise training: how do we teach recognition of patterns we ourselves do not fully understand?

One of the most effective ways to reveal subtle category differences is to transform an image and rapidly flip between the original and its altered version to highlight differences. In scientific domains, using generative models for such targeted image editing faces three key challenges: (1) automatically identifying discriminative features that may not be known or easily articulated even by experts, (2) limiting changes exclusively to these category-defining features, and (3) preserving all other identity characteristics of the instance. We develop a system that combines state-of-the-art image editing techniques with visual algebraic conditioning guidance to address these challenges in data-scarce scientific domains. Our approach automatically identifies discriminative features through visual algebraic operations that extract category-specific information without requiring explicit articulation. By integrating inverted noise maps ($z$) to preserve identity features with conditioning vectors ($c$) that guide category transformations, our system achieves effective identity-preserving yet category-changing results, that isolate and visualize subtle differences between scientific categories.

Our approach overcomes limitations in current counterfactual visualization methods, which have traditionally been applied in domains where category distinctions are already well-understood and easily verbalized. Text-guided editing methods rely on linguistic descriptions, which can be too ambiguous to specify desired visual changes. Methods like Concept Sliders (Gandikota et al., 2023), which is guided by the image distributions themselves, depend on paired examples in most cases—a constraint limiting their use in teaching scenarios. Visual counterfactual generation methods often rely on gradients from a classifier, a limitation when data is scarce. Classifier-free alternatives, like TIME (Jeanneret et al., 2024), struggle with image quality and coherence for subtle differences.

Through experiments across six domains, we demonstrate our approach's effectiveness in highlighting visual differences between categories. For instance, in black hole simulations, where distinguishing characteristics between MAD and SANE models remain largely unknown, our counterfactual visualizations emphasize distinct visual patterns in the image distribution. The transformations draw attention to variations in the uniformity of wisps and prominence of the photon ring, which are features that black hole experts themselves had not previously identified.

User studies confirm the effectiveness of our approach: participants who trained with our counterfactual visualizations demonstrated significantly better category differentiation performance than those using traditional approaches with unpaired images. This validates that our method highlights meaningful visual patterns that can be used to build expertise, even when those subtle patterns have not yet been explicitly identified or understood.

## 2 RELATED WORK

**Visual Counterfactual Explanations.** A counterfactual image shows how an input would appear if altered to switch its class, enhancing interpretability. Counterfactual inference crafts images that not only differ in classification but also clarify the visual features defining each distribution. Approaches for visual counterfactual explanations (VCEs) make use of generative model edits, with VAEs (Rodriguez et al., 2021), GANs (Lang et al., 2021), and more recently, diffusion-based methods (Jeanneret et al., 2022; 2023; 2024; Augustin et al., 2024; Sobieski & Biecek, 2024; Farid et al., 2023). Most diffusion-based approaches adapt classifier guidance (Dhariwal & Nichol, 2021) to steer the generative process of counterfactuals, requiring access to the classifier and test-time optimization to produce counterfactual images. However, generating counterfactuals this way can be challenging, as the optimization problem closely resembles that of adversarial examples. TIME (Jeanneret et al., 2024) proposes an alternative approach by using Textual Inversion (Gal et al., 2022) to encode class and dataset contexts into a set of text embeddings, providing a black-box framework for counterfactual explanations. While this removes the need for direct classifier access, Textual Inversion is primarily designed for personalization, focusing on regenerating con-

cepts in novel scenes rather than preserving image structure-an essential aspect of counterfactual generation.

**Image Editing.** Recent advances in text-to-image diffusion models (Ramesh et al., 2022; Rombach et al., 2022; Saharia et al., 2022; Nichol et al., 2022; Labs, 2024) have enabled test-time controls for image editing, ranging from semantic modifications to attention-based edits and latent space manipulation. Early approaches, such as SDEdit (Meng et al., 2022), applied noise to an image and then denoised it using a new prompt, but this often resulted in significant structural changes. Later methods refined direct prompt modifications by incorporating cross-attention manipulations or masking to better preserve image structure (Hertz et al., 2022; Parmar et al., 2023; Brack et al., 2024; Tumanyan et al., 2023; Couairon et al., 2022). Unlike single-image editing methods, Concept Sliders (Gandikota et al., 2023) introduce a different approach by optimizing a global semantic direction across the diffusion model. While text pairs can guide their optimization, they also propose visual sliders based on image pairs. However, the visual slider approach struggles with unpaired data.

**Diffusion Models with Image Prompts.** Text-to-image diffusion models generate images from text prompts, but text often falls short in capturing nuanced concepts. Image prompts offer a richer alternative, conveying nuanced details more effectively, as "a picture is worth a thousand words." DALL-E 2 (Ramesh et al., 2022) pioneered this by conditioning a diffusion decoder on CLIP image embeddings, aided by a diffusion prior for text mapping. Later works offer different architectures (Razzhigaev et al., 2023) or adapt text-to-image models for image prompts (Ye et al., 2023; Arar et al., 2023; Guo et al., 2024).

**Diffusion Inversion.** Editing a real image typically requires first obtaining a latent representation that can be fed into the model for reconstruction. This latent representation can then be modified, either directly or by altering the generative process, to produce the desired edit. Most diffusion-based inversion methods rely on the DDIM (Song et al., 2022) sampling scheme, which provides a deterministic mapping from a noise map to a generated image (Mokady et al., 2022; Wallace et al., 2022; Parmar et al., 2023). However, this approach introduces small errors at each diffusion step, which can accumulate into significant deviations, particularly when using classifier-free guidance (Ho & Salimans, 2022). Instead of predicting an initial noise map that reconstructs the image through deterministic sampling, an alternative approach considers DDPM (Ho et al., 2020) sampling and inverts the image into intermediate noise maps (Wu & la Torre, 2022). Building on this, (Huberman-Spiegelglas et al., 2024) proposed an inversion technique for the DDPM sampler, along with an edit-friendly noise space better suited for editing applications. We use this technique while conditioning on image prompts.

**Machine Teaching.** Machine teaching optimizes human learning via computational models. Early work framed this as an optimization task, minimizing example sets for efficient teaching (Zhu, 2015). Generally, the field of machine learning for discovery has machine teaching as a goal (Jumper et al., 2021; Chiquier & Vondrick, 2023). Recent advances leverage generative models and LLMs for cross-modal discovery, synthesizing representations for conceptual learning (Chiquier et al., 2024), decoding structures in mathematics, or programs for scientific discovery (Mall et al., 2025; Romera-Paredes et al., 2024). Parallel efforts amplify subtle signals for perception: language models detect fine-grained textual differences (Dunlap et al., 2024), while video motion magnification enhances visual cues (Liu et al., 2005; Wu et al., 2012; Oh et al., 2018). These methods, though effective for fine-grained discrimination, typically require aligned, abundant data and focus on single modalities. Our work extends these efforts, using diffusion models to generate visual counterfactuals for nuanced category learning.

## 3 METHOD

We begin by introducing *DIFF*usion for counterfactual image generation, as illustrated in Figure 2. In Section 3.1, we provide the necessary background on diffusion models. In Section 3.2, we present our proposed method, outlining its design and implementation.

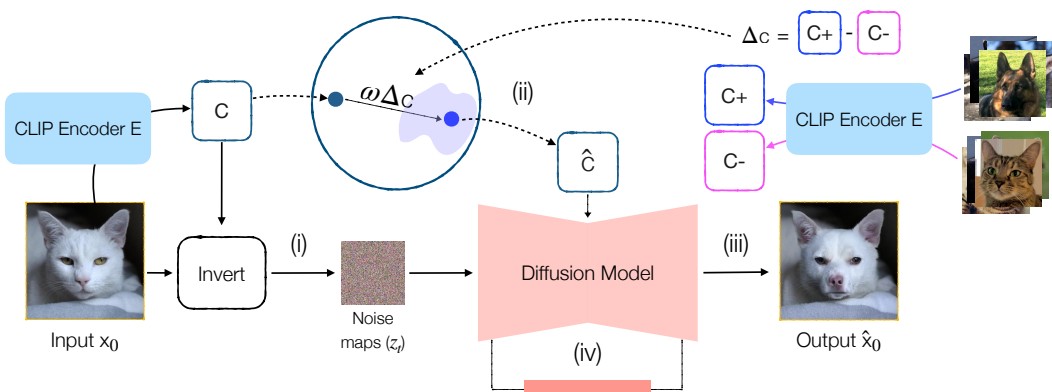

Figure 2: **DIFFusion method.** Our method consists of four parts. (i) Inverting the real image with DDPM-EF to obtain noise maps. (ii) Performing conditioning space arithmetic using positive and negative embeddings obtained from the training set. (iii) Generation via diffusion sampling, starting from the inverted noise conditioning on the manipulated conditioning vector $\hat{c}$. (iv) Optional domain tuning, in which we fine-tune the diffusion model for domain adaptation.

## 3.1 DIFFUSION PRELIMINARIES

Diffusion models generate data by sampling from a distribution through iterative denoising of noisy intermediate vectors. A forward process is first applied, where noise is gradually added to a clean image $x_0$ over $T$ steps. A noisy sample at timestep $t$ can be expressed as

$$x_t = \sqrt{\bar{\alpha}_t}x_0 + \sqrt{1 - \bar{\alpha}_t}\epsilon, \quad t = 1, ..., T \tag{1}$$

where $\epsilon \sim \mathcal{N}(0, \mathbf{I})$, $\alpha_t$ is a predetermined variance schedule, and $\bar{\alpha}_t = \prod_{i=1}^{T}\alpha_i$. The model learns to reverse the forward noising process, which can be expressed as an update step over $x_t$,

$$x_{t-1} = \mu_\theta(x_t, c) + \sigma_t z_t, \quad t = T, ..., 1 \tag{2}$$

where $z_t$ are i.i.d standard normal vectors, $\sigma_t$ is a variance schedule, and $\mu_\theta(x_t, c)$ is typically parameterized as:

$$\mu_\theta(x_t, c) = \frac{1}{\sqrt{\alpha_t}}\left(x_t - \frac{1 - \alpha_t}{\sqrt{1 - \bar{\alpha}_t}}\epsilon_\theta(x_t, t, c)\right) \tag{3}$$

Here $\epsilon_\theta(x_t, t, c)$ is the trained noise prediction network, and $c$ is an optional conditioning context, such as an image prompt embedding.

## 3.2 DIFFUSION

Given an input image $x_0$, our goal is to find a fine-grained, discriminative edit that changes a classifier's prediction. Let $\mathcal{R}_\theta(\mathbf{z}, c)$ be the recursive application of the denoising diffusion model from Equation 2. Our approach finds these edits by inverting the image $x_0$, into a sequence of noise maps, $\mathbf{z}$, and manipulating the CLIP embeddings of the original image, $c = E(x)$, into a resulting conditioning vector $\hat{c}$, before sampling the modified image. We generate the modified image $\hat{x}_0$ through:

$$\hat{x}_0 = \mathcal{R}_\theta(\mathbf{z}, \hat{c}) \tag{4}$$

Since the diffusion model must generate an image consistent with the original noise maps $\mathbf{z}$, and has a conditioning vector $\hat{c}$ that steers from the source towards the target class, the resulting samples maintain the identity of the original image, but with subtle modifications such that the class label flips.

**Inversion.** We are interested in extracting noise vectors $\mathbf{z}$, such that, if used in Equation 2, would recover the original image $x_0$. Note that any sequence of $T + 1$ images $x_0, ..., x_T$ can be used to extract consistent noise maps for reconstruction by isolating $z_t$ from Equation 2 as

$$z_t = \frac{x_{t-1} - \mu_\theta(x_t, c)}{\sigma_t}, \quad t = T, ..., 1 \tag{5}$$

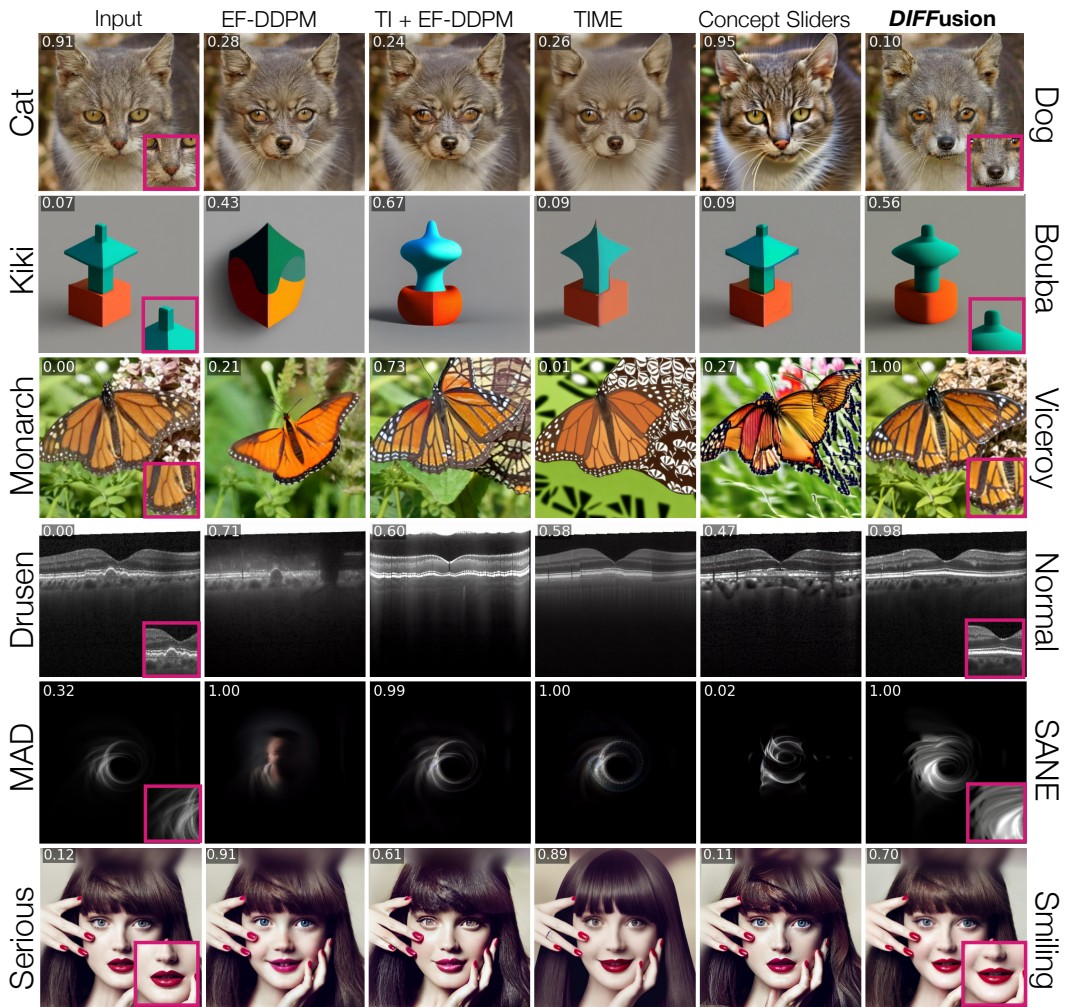

Figure 3: **Qualitative Results.** We present our qualitative results, where each row corresponds to one direction of our binary datasets. The first column contains the inputs, and each subsequent column contains the results from each baseline, with the last column containing the result from **DIFFusion**. In particular, the magnified boxes in the magenta frame show that our method is able to pick up on small discriminative cues. For example, when converting from MAD to SANE, the whisps become amplified and more uniform in brightness, and when converting from Monarch to Viceroy, a cross-sectional line is added on the wing. **Note:** The value in the top left corner of each image represents the probability predicted by the oracle classifier, as explained in Section 4.2.

We follow the choice suggested in (Huberman-Spiegelglas et al., 2024) and compute the noise maps through the standard forward diffusion process Equation 1, but using statistically independently-sampled noise for each timestep. This yields noise maps $\mathbf{z} = \{x_T, z_T, \ldots, z_1\}$ that are consistent with $x_0$.

**Conditioning.** We generate edits that flip the category through arithmetic operations on $c$, resulting in $\hat{c}$. We apply an additive translation to the conditioning vector $c = E(x)$:

$$\hat{c} = c + \omega \Delta c \tag{6}$$

where $c$ is the CLIP image embedding of the original image, $\Delta c$ is a direction that moves the class from the original class to the target class, and $\omega$ is a scaler that varies the direction's strength. We calculate this translation through the difference of means for each class:

$$\Delta c = \mathbb{E}_{x_p}\left[E(x_p)\right] - \mathbb{E}_{x_n}\left[E(x_n)\right] \tag{7}$$

such that $x_p$ is an image of class $p$ and $x_n$ is an image of class $n$ (e.g, positive and negative classes). We normalize all the image embeddings with L2 norm prior to the arithmetic.

**Sampling.** We use $\hat{c}$ as the conditioning vector for DDPM sampling, paired with the inverted noise maps, $z$, to generate the counterfactual image. As suggested in (Huberman-Spiegelglas et al., 2024), we run the generation process starting from timestep $T - T_{skip}$, where $T_{skip}$ is a parameter controlling the resemblance to the input image. Therefore, similar to Equation 2, denoting the denoised edited image at timestep $t$ as $\hat{x}_t$ we have,

$$\hat{x}_{t-1} = \mu_\theta(\hat{x}_t, \hat{c}) + \sigma_t z_t, \quad t = T - T_{skip}, ..., 1 \tag{8}$$

This approach allows us to systematically steer the image generation toward the target class by adjusting the manipulation scale $\omega$, while maintaining key structural features of the original image through $T_{skip}$. Intuitively, a larger $T_{skip}$ results in fewer denoising steps under the manipulated condition $\hat{c}$, leading to greater adherence to the input image.

**Domain Tuning** We use a pre-trained diffusion model (Shakhmatov et al., 2023) that conditions on CLIP image embeddings. When adapting to a new domain, we fine-tune the model using LoRA (Hu et al., 2021), training only its cross-attention and corresponding projection layers. As discussed in B.2, we find that domain tuning is beneficial for the Butterfly (Van Horn et al., 2018) and Retina (Kermany et al., 2018) datasets, but has minimal impact on the other datasets.

**Implementation Details.** For inversion, we adapt the edit-friendly DDPM inversion scheme (Huberman-Spiegelglas et al., 2024) to our diffusion decoder (Shakhmatov et al., 2023). Specifically, we use CFG (Ho & Salimans, 2022) in both inversion and generation. We first aim to find guidance scale parameters that achieve perfect reconstruction, and then use these guidance scales for our method. This process is further discussed in B.3. To generate counterfactuals, we manipulate the conditioning space using Equation 6, adjusting the manipulation guidance scale per dataset ($\omega = 1.0$ for AFHQ, $\omega = 2.0$ for the rest of the datasets). We then sample for $T - T_{skip}$ steps, where $T = 100$ and the choice of the $T_{skip}$ parameter is further discussed in Section 4.2.

## 4 EXPERIMENTS

### 4.1 DATASETS AND BASELINES

**Datasets.** We quantitatively benchmark on datasets from diverse domains. We also note the corresponding directions under examination for each dataset in Table 1. We evaluate on AFHQ (Choi et al., 2020), CelebaHQ (Lee et al., 2020) and KikiBouba (Alper & Averbuch-Elor, 2024) as our non-scientific datasets. We also evaluate on three scientific datasets. The first is Retina (Kermany et al., 2018), a dataset of retina cross-sections, both diseased and healthy. The second is Black Holes, which is a dataset of images taken from fluid simulations of accretion flows around a

Table 1: Datasets and their classification tasks.

| Dataset | Class 0 / Class 1 |
|---|---|
| AFHQ | Dog / Cat |
| KikiBouba | Kiki / Bouba |
| Retina | Drusen / Normal |
| Black-Holes | MAD / SANE |
| Butterfly | Monarch / Viceroy |
| CelebA-HQ | Smile / No-Smile |

black hole (Wong et al., 2022). The simulations assume general relativistic magnetohydrodynamics (GRMHD) under one of two regimes: magnetically arrested (MAD) or standard and normal evolution (SANE) (Jiang et al., 2023). Finally, we also evaluate on Monarch and Viceroy, a fine-grained species classification task. Monarch butterflies evolved to be mimics of Viceroys, and the two species are notoriously difficult to tell apart.

**Baselines.** We use TIME (Jeanneret et al., 2024) as our counterfactual baseline, and replace black-box classifier labels with ground truth labels. For editing baselines, we compare against Stable Diffusion (Rombach et al., 2022) with EF-DDPM inversion (Huberman-Spiegelglas et al., 2024) using class-name prompts. To better accommodate visual concepts, we implemented another baseline that uses Textual Inversion (Gal et al., 2022) for each class of images and then applies source and

Table 2: Performance comparison across datasets. SR = Success Ratio, LPIPS = Perceptual Distance. In **bold** are the best results, and in underline are the second-best results.

| Method | Science Datasets | | | | | | | | Regular Datasets | | | |
| --- | --- | --- | --- | --- | --- | --- | --- | --- | --- | --- | --- | --- |
| | Retina | | Butterfly | | KikiBouba | | Black-Holes | | AFHQ | | CelebA-HQ | |
| | SR↑ | LPIPS↓ | SR↑ | LPIPS↓ | SR↑ | LPIPS↓ | SR↑ | LPIPS↓ | SR↑ | LPIPS↓ | SR↑ | LPIPS↓ |
| EF-DDPM | 0.39 | 0.272 | 0.86 | 0.328 | 0.68 | 0.343 | 0.73 | 0.117 | **1.0** | **0.187** | **1.0** | **0.104** |
| TI+EF-DDPM | 0.89 | 0.330 | **1.0** | 0.289 | 0.97 | 0.332 | 0.5 | **0.045** | **1.0** | 0.211 | **1.0** | 0.181 |
| TIME | 0.50 | 0.358 | 0.13 | 0.320 | 0.17 | **0.170** | 0.52 | 0.086 | 0.95 | 0.217 | 0.79 | 0.166 |
| Concept Sliders | 0.48 | 0.248 | 0.27 | 0.362 | 0.13 | 0.206 | 0.53 | 0.155 | 0.49 | 0.375 | 0.21 | 0.238 |
| *DIFF*usion | **0.98** | **0.217** | **1.0** | **0.218** | **0.98** | 0.176 | **1.0** | 0.076 | **1.0** | 0.245 | **1.0** | 0.116 |

target prompts based on the desired edit direction. We term this baseline TI + EF-DDPM. Lastly, we use the visual sliders objective of Concept Sliders (Gandikota et al., 2023) that provides a visual counterpart to text-driven attribute edits. To ensure a robust evaluation, we experimented with varying the rank and number of images used for defining the concept direction, selecting the best configuration for each dataset. Since the original method assumes paired data, we adapted it for unpaired settings.

## 4.2 EDITING RESULTS

We quantitatively evaluate how well our method can make minimal edits to the image to flip the classifier's prediction. For evaluation, we take a balanced sample of 50 images per class from the validation set of each dataset, totaling 100 images from each dataset. Since our method can generate different strengths of edits, to pick the minimal edit, we generate 10 edits with varying strengths using the $T_{skip}$ parameter, as does the TIME baseline (Jeanneret et al., 2024), testing from highest to lowest $T_{skip}$, and select the first edit that flips the classifier prediction while maximizing LPIPS similarity to the original image.

**Metrics.** We evaluate our method using two key metrics. Success Ratio (SR): Also known as Flip-Rate, quantifies the ability of a method to flip an oracle classifier's decision. The oracle classifier we use is an ensemble of ResNet-18 (He et al., 2015), MobileNet-V2 (Sandler et al., 2019), and EfficientNet-B0 (Tan & Le, 2020), trained on each dataset. LPIPS (Zhang et al., 2018): Measures the perceptual similarity between the input and generated image, by capturing feature-level difference in a learned embedding space.

**Quantitative Results.** As seen in Table 2, our method achieves the highest SR across all datasets compared to baseline approaches. In terms of LPIPS, it shows significant improvements over previous methods on datasets where language struggles to capture visual details (e.g., Black-Holes, KikiBouba), unlike datasets with common objects like AFHQ. It also performs either best or competitively on the remaining natural-image datasets. Additionally, while TI + EF-DDPM improves the same text-based baseline, it still struggles with images that are hard to describe textually, such as Black-Holes.

**Qualitative Results.** In Figure 3, we present class transitions for all baselines and *DIFF*usion. On familiar datasets like CelebA-HQ and AFHQ, our method performs well, similar to baselines. However, its strengths stand out in datasets where language may not fully capture visual details. For KikiBouba, only our method and TI + EF-DDPM round Kiki's edges, though the baseline changes the original colors, while ours keeps them intact. In the Butterfly dataset, the baselines miss the cross-sectional line, and in the Retina dataset, only our approach removes Drusen while preserving image identity. For the Black-Holes dataset, our method flips the classifier's prediction with notable visual differences, as also highlighted in Figure 4b. These results suggest our method handles subtle visual nuances particularly well.

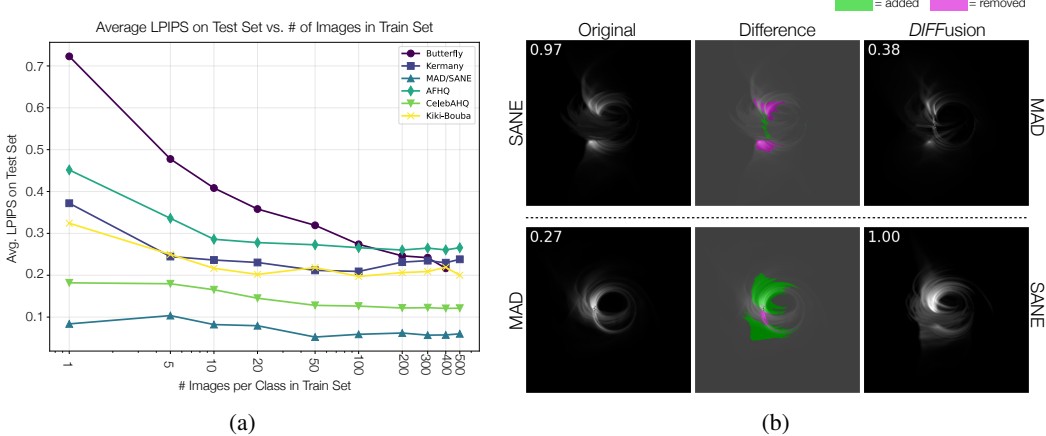

(a)                                                        (b)

Figure 4: (a) **Varying number of images.** Average LPIPS vs. number of images used per class. LPIPS stabilizes around 50 images for most datasets, reflecting improved identity fidelity and subtle class-distinctive feature shifts with increased embedding samples. (b) **Difference Overlay.** We visualize the difference between the input image and the counterfactual from *DIFF*usion. From SANE to MAD we notice a highlighting of the photon ring (**green**). From MAD to SANE we notice that the ring becomes less pronounced (**magenta**), and wisps appear (**green**).

## 4.3 TEACHING RESULTS

We evaluate our method's effectiveness in teaching people subtle visual differences between classes.

**User Study Design.** We divided participants into three groups of 10 people each. Group 1 studied only unpaired images. Group 2 studied videos transitioning from original images to counterfactual images generated by the best baseline. Group 3 studied videos transitioning from original images to counterfactual images generated by our method. Since Groups 2 and 3 viewed transitions from real to edited images, they were also exposed to the unpaired image distribution seen by Group 1. All participants studied their respective materials for 3 minutes to learn to distinguish between the two classes before taking a test. The test required labeling 50 images, evenly distributed with 25 images from each class.

**User Study Results.** We assess *DIFF*usion for teaching via a user study on the Black Holes and Butterfly datasets (Van Horn et al., 2018), shown in Table 3 and Figure 5. For Black Holes, unpaired material gave a 78% average score, but our counterfactuals boosted this to 90%, with 40% of users hitting near-perfect scores (96%+), surpassing baselines and counterfactuals. For Butterfly, unpaired data led to varied scores, but our counterfactuals raised 9 out of 10 users above 80%, standardizing understanding effectively. P-tests confirm signif-

Table 3: User Study Results - Mean Accuracy (%)

| Method | Black Holes Mean±SD | Butterfly Mean±SD | Avg. Impr. |
|---|---|---|---|
| Unpaired | 78.6±13.7 | 61.6±22.8 | — |
| Baseline | 77.2±11.5 | 62.8±16.8 | -0.1% |
| Ours | 90.8±4.8 | 87.8±10.4 | +19.2% |

icance: Black Holes ($p = 0.016$ vs. 0.811 for baseline) and Butterfly ($p = 0.004$ vs. 0.897 for baseline), both $p < 0.05$. Our counterfactuals consistently outperform alternatives, demonstrating the usefulness of our method for teaching humans subtle visual differences.

## 4.4 METHOD ANALYSIS

**Varying Dataset Size.** In Figure 4a, we examine the impact of varying the number of images per class on the average LPIPS metric across the test sets. We notice that for most datasets, the LPIPS stops improving at around 50 images. In Section B.4, we show qualitative results as the number of images changes. We notice that as the number of images incorporated into the average embeddings

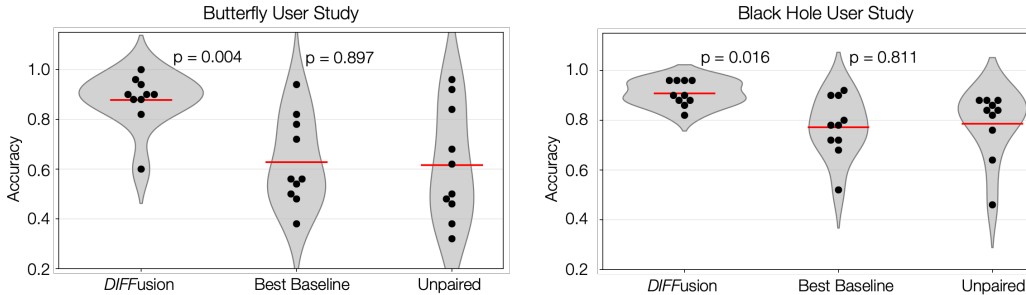

Figure 5: **User Study Results**. We plot the results from user studies across users who studied our counterfactuals, users who studied the best baseline counterfactuals, and users who studied unpaired images. For both Butterfly and Black Hole datasets, we observe that the users who studied our counterfactuals significantly outperformed the other groups. The violin plots illustrate the distribution of user percentages, where the width of each grey shape represents the density of data points.

increases, the fidelity to the original image's identity improves, while subtly altering the features that are distinctive between classes.

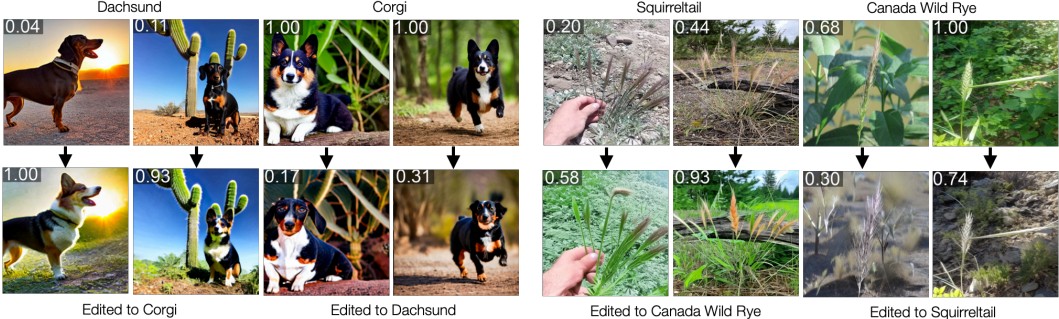

Figure 6: **Dataset Bias**. *DIFF*usion reveals dataset bias. Squirreltail-to-Canada Wild Rye edits mphasize environmental backgrounds over plant traits, reflecting iNaturalist's contextual bias, and Dachshund-to-Corgi edits prioritize foreground dog features, yet still reflect environmental bias.

### 4.5 VISUALIZING DATASET BIAS

Our method edits images using differences between class mean embeddings, making it sensitive to dataset bias. If distinguishing features reflect unintended biases rather than targeted traits, edits deviate from our intent. This is both a limitation - preventing precise control, and a strength, as it visualizes dataset biases, revealing underlying structure. We show how dataset bias is captured by our method in Figure 6. In iNaturalist (Van Horn et al., 2018), counterfactuals from Squirreltail (dry climates) to Canada Wild Rye (humid) shift backgrounds more than plant structure, suggesting environmental bias dominates. Conversely, using the Spawrious (Lynch et al., 2023) dataset, Dachshund-to-Corgi counterfactuals prioritize dog features (e.g., shape, size) over jungle-to-desert backgrounds. We attribute this to stronger foreground differences in dogs and clearer object-background separation, unlike plants blending into settings in iNaturalist data. The effect of dataset bias on edits varies with class prominence and context.

## 5 DISCUSSION AND LIMITATIONS

*DIFF*usion generates counterfactuals to support visual expertise training across domains with limited data. It reveals dataset biases, often shifting unintended features due to embedding reliance, which limits precise control. Additionally, the arithmetic is very simple: a difference of averages, highlighting a trade-off between flexibility and specificity. Future work could explore disentanglement or guidance mechanisms to enhance edit precision in specialized applications.

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
