# A IMPLEMENTATION DETAILS

## A.1 BASELINES IMPLEMENTATION DETAILS

As described in Section 4.1, we compare our method against the following baselines: TIME (Jeanneret et al., 2024), Stable Diffusion (Rombach et al., 2022) with EF-DDPM inversion (Huberman-Spiegelglas et al., 2024), once using class names as text prompts, and once using learned textual embeddings of a group of each class' images through Textual Inversion (Gal et al., 2022). Lastly we also compare to Concept Sliders (Gandikota et al., 2023). We used the following third-party implementation in this project:

- TIME: (Jeanneret et al., 2024). Instead of using their classifier's labels to group the data into two classes, we used the ground-truth labels. Additionally, to perform the evaluations, we used our own ensemble classifiers. official implementation.

- EF-DDPM (Huberman-Spiegelglas et al., 2024): official implementation. For the prompt-based baseline, we use class names as prompts of the form "*a photo of a CLASS-NAME*", where CLASS-NAME can be *"cat", "viceroy", "drusen"*, etc. For the scientific datasets we also include an identifier of the form *"Butterfly", "Black Hole", "Retina"*. For the textual-inversion-based baseline (TI + EF-DDPM), we first invert each class of images into a newly added token and save when optimization is done. Then, we use the source-class token as the inversion prompt, and the target-class token as the generation prompt.

- Concept Sliders (Gandikota et al., 2023): official implementation. We built on the official implementation of official implementation, utilizing their Visual Concept Sliders and image editing script. While their approach assumes paired data, we found that unpaired data performs well on datasets with well-aligned classes, such as AFHQ. Conversely, for datasets with less alignment, like Butterfly, training with fewer images, ranging from 5 to 20, slightly improved results. To optimize performance, we varied both the number of training images and the LoRA rank for each dataset, evaluating combinations to select the best one per dataset.

As discussed in Section 4.2, our evaluation algorithm starts by adjusting parameters that have the strongest manipulation effect first (e.g. guidance scales, $T_{skip}$, and manipulation scale), and stops as soon as a flip occurs, logging the generated image along with the parameters that caused the flip. If no flip is achieved, we apply the final set of parameters designed to induce the most significant edit.

## A.2 INTERPOLATION

In Figure 7, we present qualitative results demonstrating the effects of varying the manipulation scale, $w$, on an instance of a Normal retina. The manipulation scale, which can take positive or negative values, modulates the transformation direction. Positive values of $w$ shift the features toward Drusen from the Normal retina, while negative values make the image smoother.

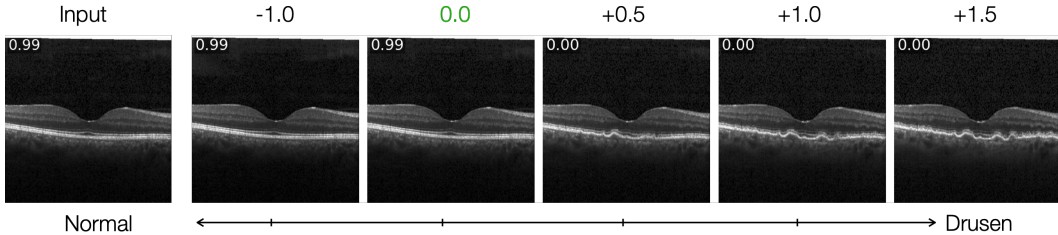

Figure 7: **Interpolation.** By varying the manipulation scale $\omega \in \{-0.5, 0.0, 0.5, 1.0, 1.5\}$, we can adjust the manipulation strength, allowing for smooth interpolation between the two classes. When $\omega = 0$, we can reconstruct the image while preserving the classifier's probabilities.

### A.3 USER STUDY DETAILS

Group 1 studied a folder of unpaired images for 3 minutes. This folder contained images of both classes, with the label of the class written on top of the image. Group 2 studied the unpaired image folder for 1 minute, the counterfactuals generated by the best baseline from Class 0 to Class 1 for another minute, and finally the counterfactuals generated by the best baseline from Class 1 to class 0 for a minute. Group 3 followed the same protocol as Group 2, except with our counterfactuals instead. For both Butterfly and Black hole, the best baseline was TI + DDPM-EF, according to LPIPS. We only showed counterfactuals where the class flipped.

The participants of the user studies were undergraduates and graduates who volunteered in exchange for baked goods. The supplementary material contains videos of the study material for Group 3, for both the black hole and the butterfly dataset. No user had any prior knowledge about either of the datasets before studying the material and taking the test.

### A.4 TRAINING DETAILS

We utilize the diffusion decoder from (Shakhmatov et al., 2023) and optionally fine-tune cross-attention weights using LoRA (Hu et al., 2021) on either subsets or the full dataset. For fine-tuning, we set the LoRA rank to $4$, the LoRA scaling factor to $\alpha = 8$ and use a base learning rate of $0.003$. Fine-tuning was conducted on a single NVIDIA A100 GPU, although the implementation supports multi-GPU training as well. We train for $4$ epochs, and select the checkpoint with the optimal balance between LPIPS and classifier accuracy on generated counterfactuals as our final model for each dataset (or data subset).

### A.5 DATASETS

- AFHQ (Choi et al., 2020): official implementation. Creative Commons BY-NC 4.0.
- KikiBouba (Alper & Averbuch-Elor, 2024): official implementation. MIT License.
- Retina (Kermany et al., 2018): official implementation. Creative Commons BY-NC 4.0.
- Butterfly (Van Horn et al., 2018): official implementation. MIT License.
- CelebA-HQ (Lee et al., 2020): official implementation. Creative Commons BY-NC 4.0.

## B   ADDITIONAL EXPERIMENTS

### B.1   SR VS. LPIPS CURVES

In Figure 8 we plot the Success Ratio vs. LPIPS curves for our method compared to the best baseline - TI + EF-DDPM, rather than choosing a single set of parameters which is required to report Table 2. Since both use the same inversion technique, we create these curves by varying the $T_{skip}$ parameter. A higher AUC generally indicates a better tradeoff between classifier flip-rate and similarity to input images for each dataset. Our method outperforms the best baseline across all datasets while achieving comparable performance on AFHQ.

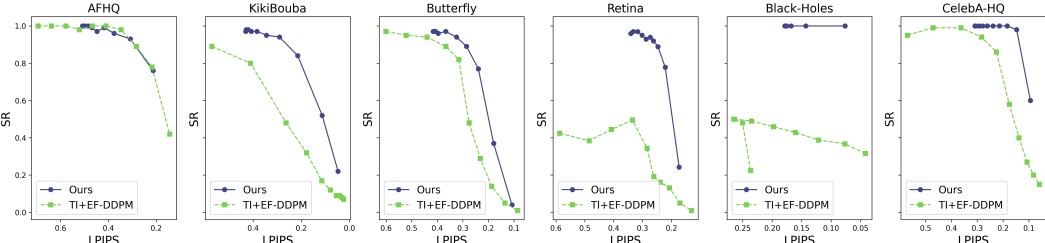

Figure 8: **Success Ratio (SR) vs. LPIPS curves.** As discussed in Section B.1, we fix the guidance scales for each method, and the manipulation scale $\omega$ for ours according to the implementation details in Section 3.2. We then vary $T_{skip}$ in increments of 0.1 within the range $[0.0, 0.9]$, where $T_{skip}$ represents the percentage of timesteps skipped relative to the total denoising steps.

### B.2   RESULTS W/O DOMAIN TUNING

As discussed in Section 3.2, we begin by fine-tuning a LoRA adapter (Hu et al., 2021), applied to the cross-attention and their linear projection weights, using a simple loss at random timesteps $t$, i.e:

$$\mathcal{L}_{simple} = \mathbb{E}_{x,\epsilon,t} \left[ \|\epsilon - \epsilon_\theta(x_t, t, c)\|_2^2 \right]$$

where the prompts $c$ are derived from image embeddings of training set examples. To obtain the best weights, we log SR and LPIPS scores on a small validation set at the end of each epoch. In certain datasets, fine-tuning has minimal to no impact on the overall results. This suggests that, in some cases, the prior learned by the pre-trained diffusion model is sufficiently strong to produce meaningful edits when conditioned on a manipulated CLIP image embedding. The results of this analysis are presented in Table 4, where we observe that domain tuning plays a crucial role in datasets like Retina and Butterfly, while having a lesser effect on others.

### B.3   PERFECT INVERSION

Perfect reconstruction can achieved when the same conditioning prompt is used during both inversion and sampling. In this case, we hope that the original image is fully reconstructed. However, DDIM (Song et al., 2022) introduces small errors at each timestep, making exact reconstruction challenging, especially with a limited number of timesteps or within the classifier-free guidance framework (Ho & Salimans, 2022). Recent works (Huberman-Spiegelglas et al., 2024; Wu & la Torre, 2022; Brack et al., 2024) focus on non-deterministic DDPM inversion and have demonstrated perfect image reconstruction when applied to Stable Diffusion (Rombach et al., 2022). Since we are

Table 4: **Results without Domain Tuning.** We evaluate our method without fine-tuning on each dataset, measuring performance using Success Ratio (SR) and perceptual distance (LPIPS). Compared to Table 2, we observe substantial improvements in both SR and similarity for the Retina and Butterfly datasets, as well as noticeable gains in reconstruction for Black-Holes and KikiBouba. However, the impact is minimal on the most natural-image datasets, AFHQ and CelebA-HQ. This suggests that the prior learned by the pre-trained diffusion model is strong enough to generate meaningful edits for these datasets without additional fine-tuning.

| Method | AFHQ | | KikiBouba | | Retina | | Black-Holes | | Butterfly | | Celeba-HQ-Smile | |
|---|---|---|---|---|---|---|---|---|---|---|---|---|
| | SR ↑ | LPIPS ↓ | SR ↑ | LPIPS ↓ | SR ↑ | LPIPS ↓ | SR ↑ | LPIPS ↓ | SR ↑ | LPIPS ↓ | SR ↑ | LPIPS ↓ |
| Ours | 1.0 | 0.249 | 0.98 | 0.2014 | 0.515 | 0.454 | 0.980 | 0.119 | 0.31 | 0.344 | 1.0 | 0.123 |

using an image-conditioned diffusion decoder from the Kandinsky model family (Razzhigaev et al., 2023), we first explore the choice of guidance scales required to achieve perfect reconstruction while using CFG (Ho & Salimans, 2022) in both inversion and generation. While perfect reconstruction does not necessarily guarantee a useful editing space, poor reconstruction from the start is likely to cause significant deviations from the source image, which is an undesirable outcome when generating counterfactuals. Figure 9 illustrates this effect, showing that using equal guidance terms in inversion and sampling results in good reconstruction, which starts to degrade when inversion guidance scale, $\omega_{src}$, and target guidance scale, $\omega_{tar}$, are larger than 4.

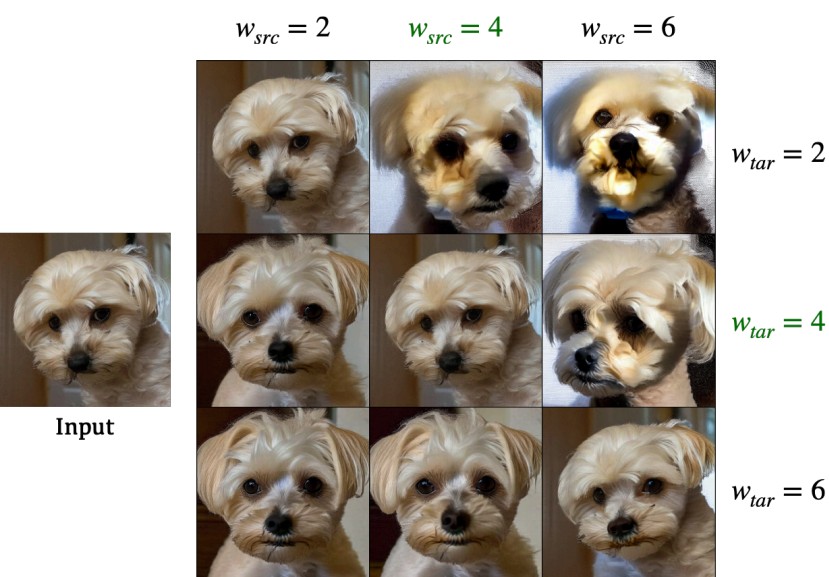

Figure 9: **Perfect Inversion.** We choose our inversion and sampling guidance scales by first reconstructing the original image with CFG. Then, we use these guidance scales for steering.

## B.4 VARYING DATASET SIZE RESULTS

As described in Section 4.4, and in Figure 4a, we demonstrate the effect of varying the number of images we have access to for applying *DIFF*usion. In this section, we show examples of generated counterfactuals per number of images in access, $N$, as shown in Figures 10,11,12,13,14. We show the grids with increasing number of images so long as the results continue to improve.

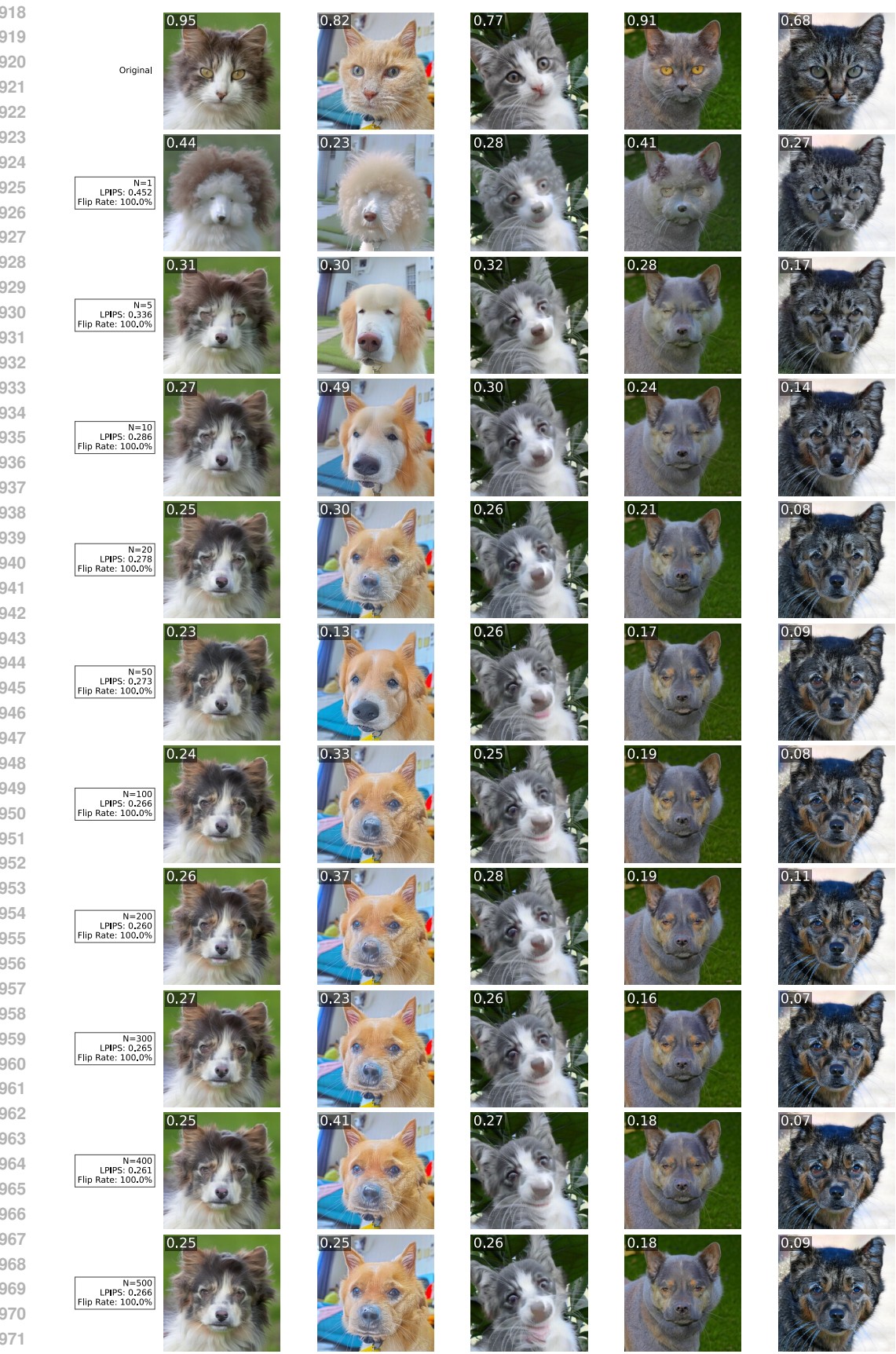

Figure 10: Varying Number of Images for AFHQ (Choi et al., 2020).

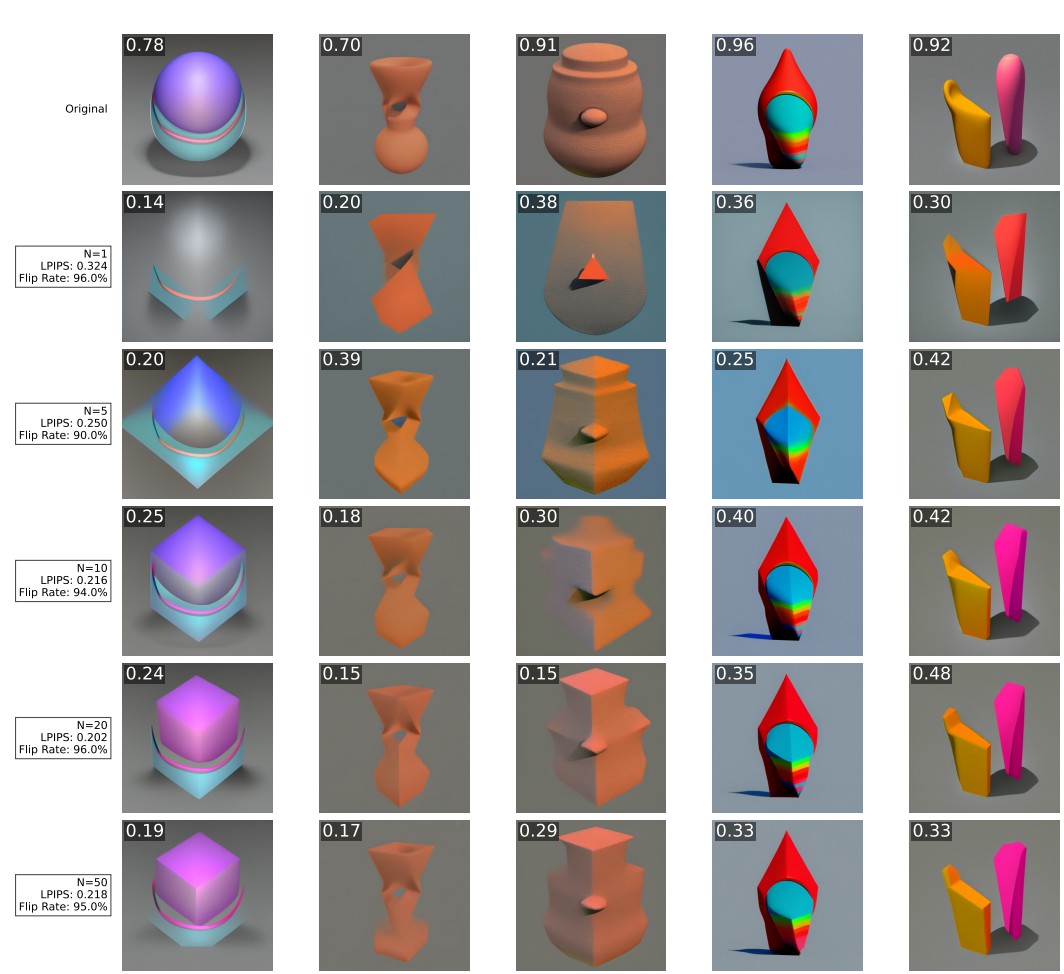

Figure 11: Varying Number of Images for KikiBouba (Alper & Averbuch-Elor, 2024).

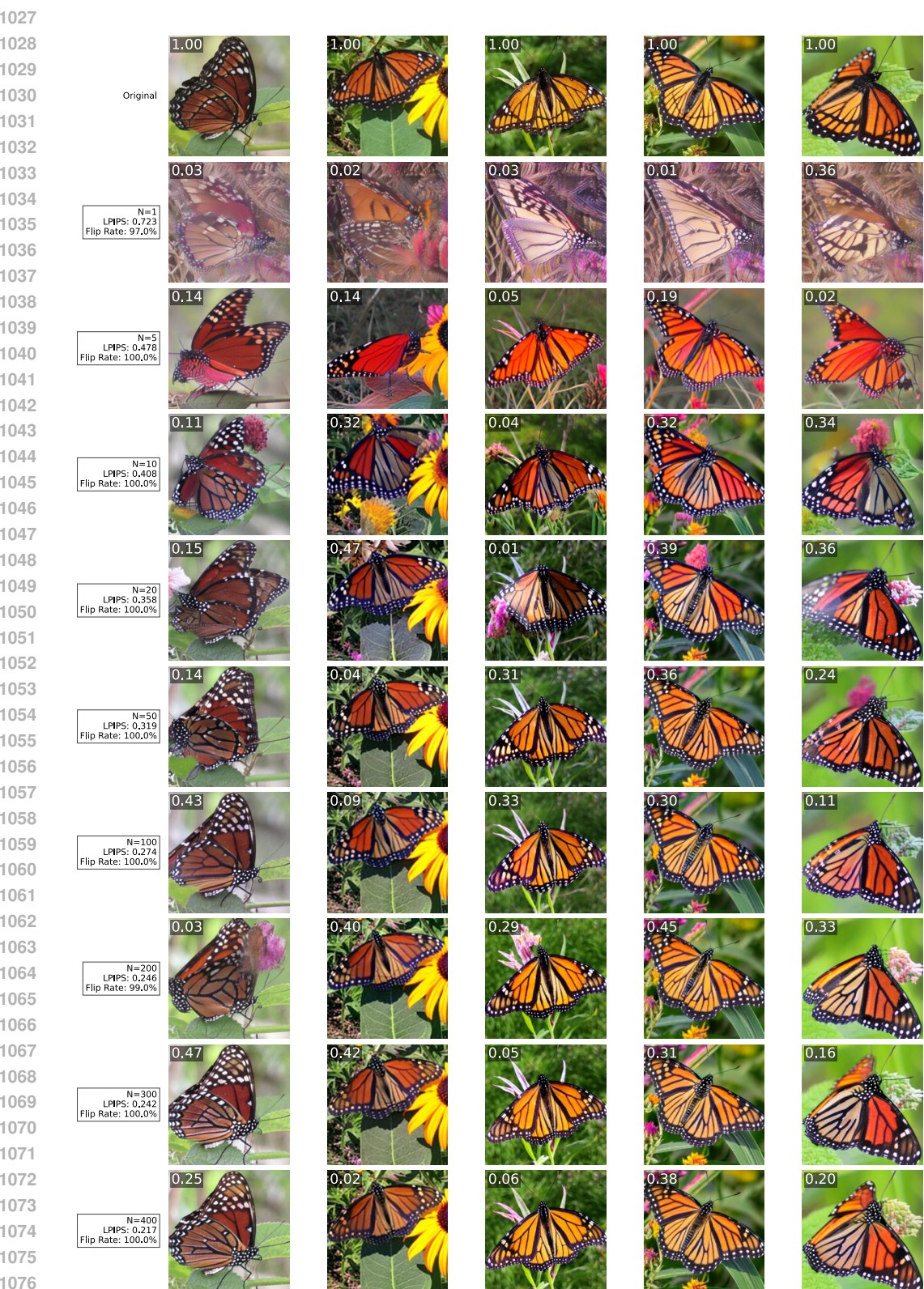

Figure 12: Varying Number of Images for Butterfly (Van Horn et al., 2018).

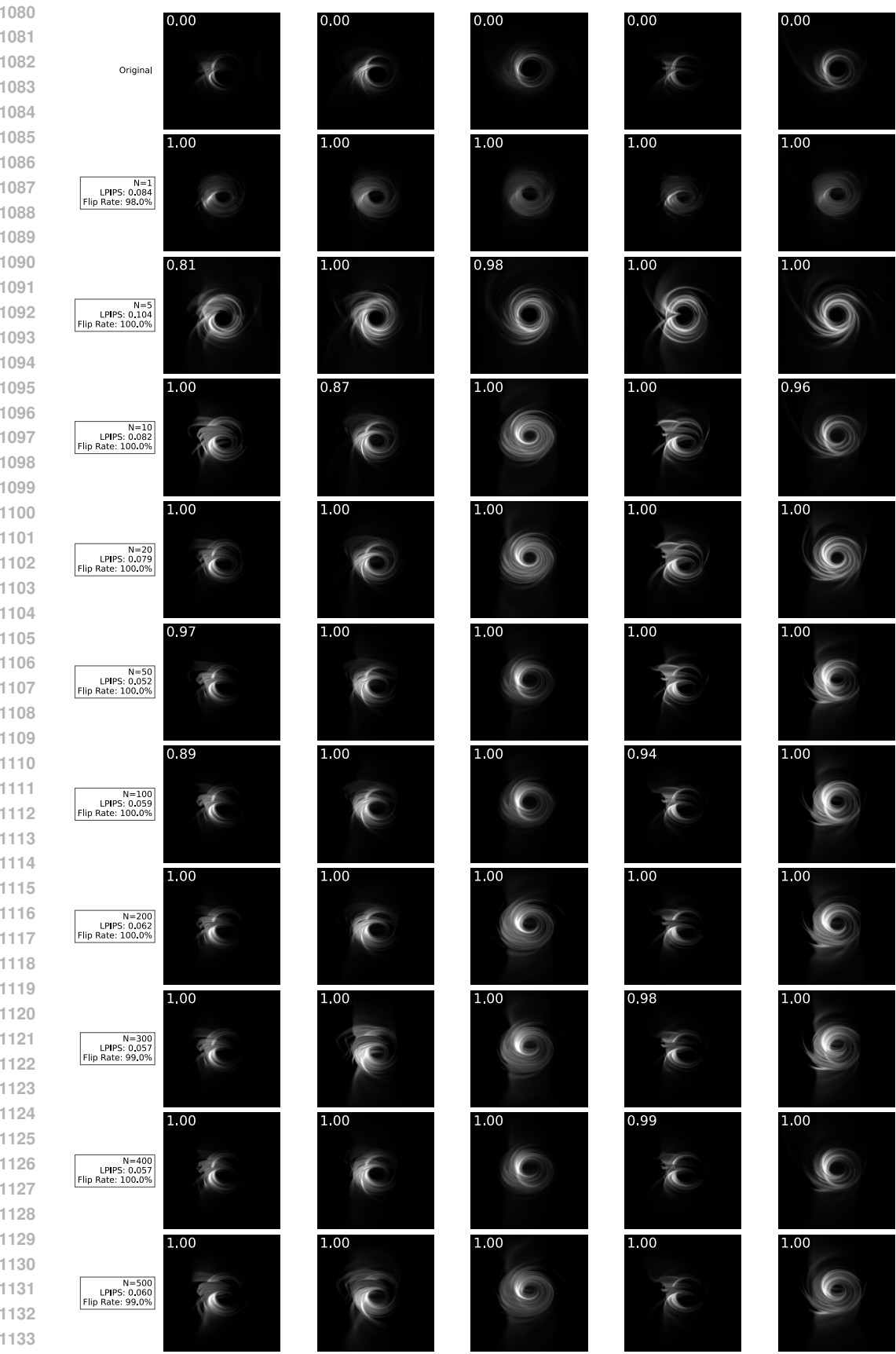

Figure 13: Varying Number of Images for Black-Holes.

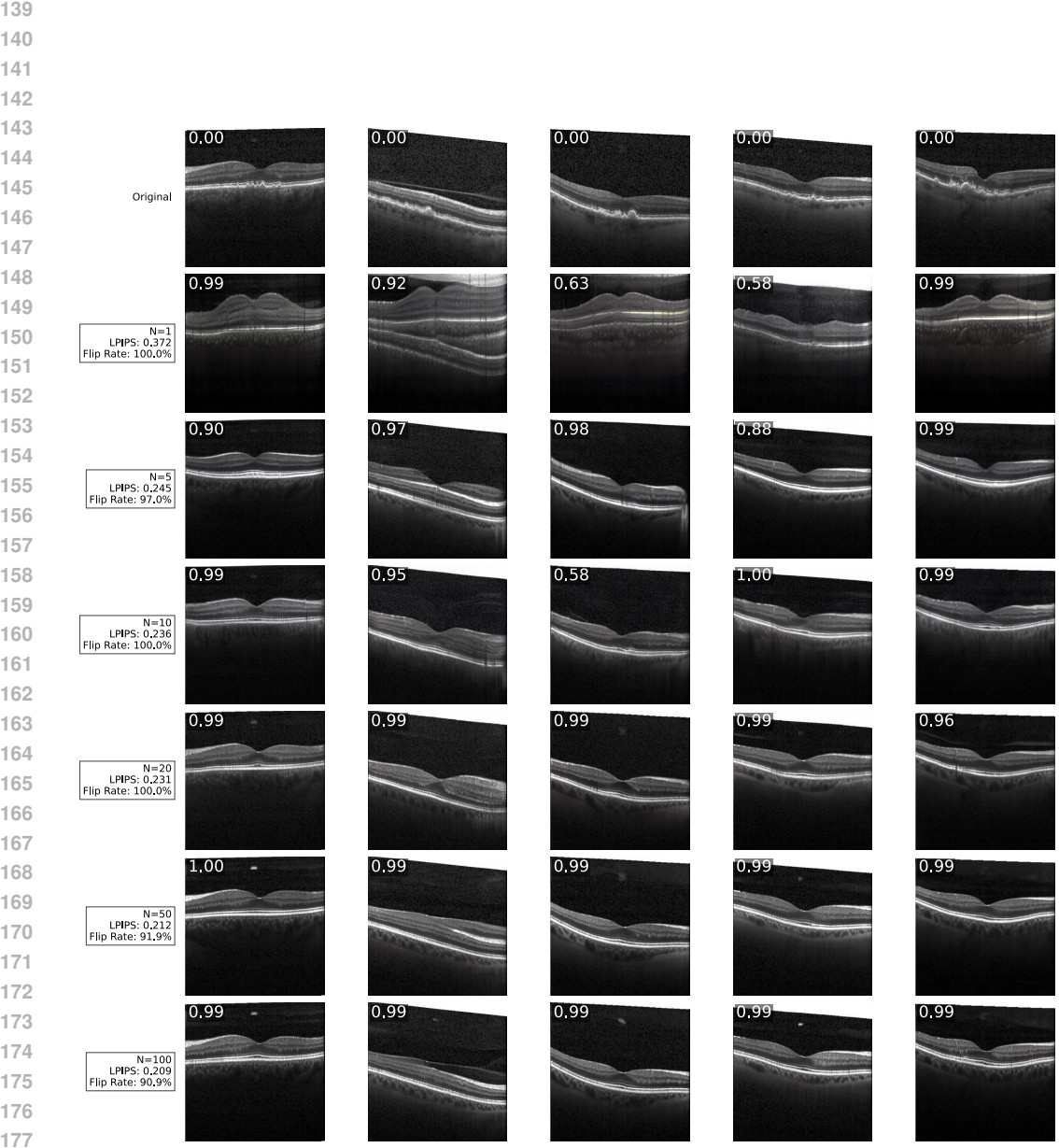

Figure 14: Varying Number of Images for Retina (Kermany et al., 2018).