# OpenReview forum: "Teaching Humans Subtle Differences with DIFFusion"
_ICLR.cc/2026/Conference — ICLR 2026 Conference Withdrawn Submission_

### Official Review · Reviewer_THd6 · 2025-10-20

**Soundness:** 3
**Presentation:** 3
**Contribution:** 2
**Rating:** 6
**Confidence:** 4

**Summary:**

The paper presents DIFFusion, a method that leverages diffusion models to automatically discover and visualise minimal discriminative features between visual categories, to teach humans to recognise subtle differences. The approach is particularly tailored for domains where text descriptions are inadequate or unavailable, such as scientific imaging, black hole simulations and fine-grained species classification.

DIFFusion inverts a real image into noise maps, manipulates the CLIP embedding by adding a class-difference vector and re-generates the image to produce identity-preserving counterfactuals that highlight only discriminative features.

**Strengths:**

The introduced methodology combines existing techniques in a novel way while maintaining simplicity.

The results indicate that the method works well across six binary classification datasets spanning both scientific and regular domains.

 This work could have a meaningful impact on scientific education and training, particularly in domains where visual expertise is crucial but difficult to teach through traditional methods.

The paper includes thorough quantitative metrics, qualitative comparisons across multiple baselines, ablation studies on dataset size and human user studies demonstrating practical teaching effectiveness.

**Weaknesses:**

The conditioning manipulation relies on simple arithmetic, which raises questions about whether this approach can capture more complex, multi-modal differences between classes or whether it may oversimplify the discriminative features.

While the paper claims to discover features unknown to domain experts, the validation of these discoveries relies primarily on classifier flip rates rather than expert verification or physical/scientific validity. Additional expert validation would strengthen these claims.

The method is susceptible to dataset bias, causing unintended feature changes (e.g., background shifts in plant images), which limits precise control despite being useful for revealing biases.

**Questions:**

Could you please elaborate on the core technical novelty of DIFFusion, distinguishing its contributions from existing state-of-the-art image editing and counterfactual generation methods?

Have you explored more sophisticated methods for computing the class-difference vector beyond the simple difference of means?

How can this methodology be extended to take on multi-class problems?

---

### Official Review · Reviewer_MFDi · 2025-10-28

**Soundness:** 2
**Presentation:** 3
**Contribution:** 3
**Rating:** 4
**Confidence:** 3

**Summary:**

Authors propose a method to generate counterfactual images. They specialize in classes where text descriptions of discriminating features are not available, which is the case in many scientific fields. To do this, they encode a target image with CLIP, and perturb it by the average of the target class - the average of the original class. They then invert the original image and condition its re-generation on the perturbed CLIP embedding, resulting in an edit in the direction of the target class.

They evaluate this on two general and four scientific examples, to see that the model incorrectly classifies the perturbed image and it stays close to the original. They also evaluate if these images can be used to teach humans to understand differences. They include additional ablations on the dataset size and biases in the datasets.

**Strengths:**

S1) The problem statement has clear applications in real-world scenarios, specifically for scientific discovery.

S2) The method is elegant.

S3) Presented results look promising.

**Weaknesses:**

My strongest concerns lie in the relationship to existing literature (see W1, W3, W4). If I were convinced that this work is well-situated within current work, and that the results are significant and baselines sufficiently encompass SOTA, I would view the work more favorably.

W1) The work does not seem well-placed within the existing literature, specifically in the areas of Visual Counterfactual Explanations, Image Editing, Diffusion Models with Image Prompts, and Diffusion Inversion. As a particular example, the section Visual Counterfactual Explanations references less than 10 examples of counterfactual work--a field which is much larger (as evidenced in [A], which may provide some starting place).

W2) The choice of text-to-image model is largely overlooked. Given the importance of the choice of model towards results, some discussion would greatly strengthen the reader's ability to understand the method's potentials and limitations (especially because it is only demonstrated on one model).

W3) Given that each dataset only includes a single category pair, 6 examples seems very little towards understanding method performance (see Q1).

W4) The number of SOTA comparison methods seems too few, especially given that they are spread over counterfactuals, text-driven editing, and vision-driven editing. E.g. TIME is the only counterfactual baseline, despite more being present in the related works (e.g. DiG-IN). As it stands, it is not clear if the presented methods fully encompass SOTA; comparing to more methods would improve the readers ability to compare this work to previous.


[A] Melistas et al., Benchmarking Counterfactual Image Generation, NeurIPS 2024.

**Questions:**

Q1) Is there justification for the dataset selections? (e.g. is it consistent with previous work)

Q2) The oracle classifiers (ResNet-18, MobileNet-V2, EfficientNet-B0) seem at first glance seem like they may not be sufficiently accurate. It would be helpful if you could provide some justification for the use of these models, e.g. statistics on their accuracy, justification from previous literature, etc. Otherwise, it may be that stronger models available could remove noise from the inaccuracy of weaker models.

---

### Official Review · Reviewer_5Seu · 2025-11-01

**Soundness:** 3
**Presentation:** 3
**Contribution:** 2
**Rating:** 4
**Confidence:** 4

**Summary:**

The paper introduces a diffusion based pipeline to generate visual counterfactual explanations (VCEs) with the goal of teaching people to correctly differentiate between fine-grained classes.

Their pipeline consists of three steps: For a given image, they (i) extract the noise vectors using inversion, (ii) compute a direction in the embedding of a CLIP vision encoder using positive and negative class samples, (iii) generate an image based on the noise vectors and conditioned on the CLIP embedding which was shifted in the computed direction. Optionally, the diffusion model is fine-tuned for domain adaption.

The pipeline is evaluated on six class pairs from six different datasets and compared to several counterfactual methods which do not utilise a classifier. The reported metrics are the flip-rate and LPIPS distance between original and edited image. Further, the authors conducted a user study to evaluate the applicability of their method to teach humans the fine-grained category differences.

**Strengths:**

The paper presents an interesting application of VCEs and provides evidence that their method produces useful visualizations for the presented examples. Further, their method has less computational cost compared to optimization methods for diffusion-based VCEs.

**Weaknesses:**

The introduced method utilises the CLIP vision encoder indirectly as a few-shot classifier (as well as the oracle classifier according to App. A.4). This few-shot classifier could be used in other classifier-guided diffusion approaches to generate VCEs for comparison. Alternatively, one could train another classifier for that purpose on the same training examples. Nevertheless, the authors should consider to add comparisons to such classifier-guided approaches as they are comparable to their method and provide stronger baselines.

Another weakness is the limited selection of examples, i.e. one class-pair per dataset. While these show interesting use cases, a broader evaluation would be required to better access the performance of the proposed pipeline.

In the context of the goal to teach humans, an open question seems to be how to distinguish true visual differences from shortcuts or other failure modes of the CLIP embedding. Also, it seems not clear whether the resulting image being out-of-distribution. e.g. by showing an object which is a composition of partial features of both classes, could have problematic side-effects on the task.

**Questions:**

1. How did the authors choose the selection of class pairs?

2. Do the generated images actually actually belong to the target class and how is this process influenced by biases/failure modes of the i) CLIP envoder ii) oracle classifier?

---

### Official Review · Reviewer_Q4m4 · 2025-11-01

**Soundness:** 2
**Presentation:** 2
**Contribution:** 2
**Rating:** 2
**Confidence:** 4

**Summary:**

The paper proposes a diffusion model method that reveals subtle, discriminative visual features between fine-grained classes through minimal, identity-preserving counterfactual images. The method works by partially inverting the image and then denoising it with an adjusted textual guidance towards the target class. Its efficacy is tested on different domains, such as black holes, butterflies, and medical images, and perform well compared to previous approaches in teaching human experts to distinguish fine-grained classes.

**Strengths:**

- Counterfactual explanations are an important research direction and using them to teach humans about fine-grained details between classes is an very relevant application.
- The method is simple, while at the same time the chosen qualitative results shows good results.
- On the evaluated datasets, the proposed method performs on par or better on success rate. The user study is very interesting and shows the method's benefits in practice.

**Weaknesses:**

- The technical novelty is limited.
  * There have been previous methods that have proposed very similar approaches of renoising and denoising to obtain slight variations of the same image. DiME [A] uses a similar idea for counterfactual generation, and in data augmentation [B] the same ideas have been applied.
  * The method combines simple primitives and previous methods and does not ablate any design choices, e.g., about the CLIP guidance or inversion process.

- The scientific writing can be improved.
  * The introduction does not sufficiently support the claims. For instance, the examples given in line 53, for "In specialized scientific domains, the complete set of visual features distinguishing between categories may be partially or entirely undiscovered", or specifically in line 83, "The transformations draw attention to variations in the uniformity of wisps and prominence of the photon ring, which are features that black hole experts themselves had not previously identified". There is no reference that these claims are true.
  * The contributions of the paper are not clear. The introduction does not clearly indicate what is novel about this paper.

- Experimental evaluation is not rigorous enough.
  * The experimental evaluation is quite narrow, only testing for 6 counterfactual class pairs.
  * There are not enough comparisons to counterfactual diffusion model methods, e.g., DVCE [C], DiME [A], and/or ACE [D].
  * The proposed method likely benefits from the LoRA fine-tuning compared to the competitors creating unfair comparisons and potentially inflating the improvement compared to baselines.
  * The dataset bias experiment is weak. There are only few qualitative results, and it is not clear if the bias comes from the diffusion model, the CLIP model, or if it can also be caused by the proposed method.

- The method seems to require careful hyperparameter choices. Hyperparameters vary per dataset and are manually chosen (to my understanding).

[A] Jeanneret et al., Diffusion Models for Counterfactual Explanations, ACCV 2022
[B] Costa et al., Diversified in-domain synthesis with efficient fine-tuning for few-shot classification, arxiv 2023
[C] Augustin et al., Diffusion Visual Counterfactual Explanations, NeurIPS 2022
[D] Jeanneret et al., Adversarial Counterfactual Visual Explanations, CVPR 2023

**Questions:**

Please refer to the weaknesses section. Questions of particular interest are:
- What are the paper's contributions?
- Do you also use LoRA fine-tuning for the competitor approaches? If not, how would the results change if you do?
- Can you add comparisons with DVCE, DiME, and/or ACE?
- Are hyperparameters chosen based on a held-out validation set?

---

### Note · Authors · 2025-12-04

**Comment:**

We thank the reviewers for their time and we appreciate the comments.

**Withdrawal Confirmation:**

I have read and agree with the venue's withdrawal policy on behalf of myself and my co-authors.